# The Importance of Resilience and Level of Anxiety in the Process of Making a Decision about SARS-CoV-2 Vaccination

**DOI:** 10.3390/ijerph20020999

**Published:** 2023-01-05

**Authors:** Natalia Maja Józefacka, Robert Podstawski, Wiktor Potoczny, Andrzej Pomianowski, Mateusz Franciszek Kołek, Sylwia Wrona, Konrad Guzowski

**Affiliations:** 1Institute of Psychology, Pedagogical University of Krakow, Podchorążych 2, 30-084 Krakow, Poland; 2Department of Tourism, Recreation and Ecology, University of Warmia and Mazury in Olsztyn, 10-957 Olsztyn, Poland; 3Department of Internal Diseases with Clinic, University of Warmia and Mazury in Olsztyn, 10-719 Olsztyn, Poland; 4Diplomstudium Humanmedizin, Medizinische Universität Wien, Spitalgasse 23, 1090 Wien, Austria; 5Faculty of Arts and Educational Science, University of Silesia in Katowice, ul. Bankowa 12, 40-007 Katowice, Poland; 6Students Scientific Club ControlUP, Institute of Psychology, Pedagogical University of Krakow, Podchorążych 2, 30-084 Krakow, Poland

**Keywords:** COVID-19 fear, anxiety, resilience, self-esteem, COVID-19 vaccination, vaccine hesitancy, pandemic

## Abstract

People’s opinions on immunization are diverse. Despite the constant improvement of vaccine formulas, the number of people reluctant to immunize is not decreasing. The purpose of our study is to assess the psychological determinants of immunization reluctance in depth. We measured levels of anxiety (death-related and general), fear of COVID-19, self-esteem and resilience among 342 adults. We found that the level of COVID-19 related fear is higher among the vaccinated population, despite general anxiety levels being lower. Surprisingly we didn’t find significant differences in resilience and self-esteem levels. Findings are concurrent with previous research—COVID-19 related fear level is higher among vaccinated people. Resilience and self-esteem are defined as stable, trait-like constructs, and thus may not manifest higher levels in very specific pandemic situations, although they may lower the levels of general anxiety.

## 1. Introduction

Vaccines have been a huge step for public health, and have greatly reduced the morbidity and mortality of many diseases [1]. Hesitation in making the decision to vaccinate is a complex problem; the WHO Strategic Advisory Group of Experts (SAGE) clarify that ‘delay in acceptance or refusal of vaccination despite the availability of vaccination services. Vaccine hesitancy is complex and context-specific, varying across time, place, and vaccines. It is influenced by factors such as complacency (e.g., a perceived need for the vaccine), convenience (e.g., accessibility of the vaccine), and confidence (e.g., perceived benefits and safety of the vaccine)’ [2]. COVID-19 vaccines were produced quickly, which raised public doubts about their effectiveness against the pandemic [3]. Despite the continuation of public vaccination promotion campaigns, only a slight increase in the willingness to accept the COVID-19 vaccine has been observed. Based on research conducted in 26 European countries, around 26% of participants declined vaccination against COVID-19 [4]. Across the world, the percentage of people who fear complications or question the effectiveness of vaccines is not decreasing significantly [3,5]. Exploring the psychological determinants of vaccine rejection could impact the creation of tailored information, and help combat the COVID-19 pandemic.

Reports made before the outbreak of the COVID-19 pandemic showed that Polish society did not have a negative attitude towards immunization [6]. Unfortunately, opinions about vaccination are becoming increasingly diverse. Among the reasons for negative COVID-19 vaccination attitudes, the following have thus far been identified: lack of trust in vaccines, fear of side effects, pharmaceutical conspiracies, and preference for a natural lifestyle [7]. The most controversial issues in the widespread use of vaccination have been the safety of the substances used in production and the number of preparations administered at an early age [8]. Additionally, global survey respondents (from 60 nations) reporting lower levels of trust in information from government sources were less likely to accept a COVID-19 vaccination [9].

According to the psychological, social, and situational factors described so far, it is important to understand the mechanisms responsible for vaccination intentions in the new context of the COVID-19 pandemic. Even before the first formulas were on the market, 25% of people would have refused the vaccine [10]. Previous studies focusing on finding predictors of vaccine readiness during the COVID-19 pandemic show that willingness to receive the vaccine is positively associated primarily with education, economic status, and risk of infection [5,11,12].

Current research is focused on better understanding the psychological variables that may account for the lack of the decision to vaccinate against SARS-CoV-2. Previous studies investigated the psychological drivers of COVID-19 vaccination intention. The proposed model includes five antecedents of vaccination: confidence, complacency, constraints, calculation, and collective responsibility (5C model). The results show that all five components of the 5C model are related to COVID-19 vaccination. However, confidence and collective responsibility are most strongly related to vaccination intention [13]. Considering the broader context of the issue, we can relate this model to psychological resilience. Resilience is responsible for positive adaptation and is required to respond to different adversities, ranging from daily troubles to critical life events [14]. Psychological resilience is an essential resource which helps to make difficult and complex decisions [15]. Therefore, it can be important when a new type of vaccine is released.

The level of anxiety could be another important variable that can influence vaccination decisions. Overall, concern about the side effects and general safety of vaccinations is one of the main factors determining vaccination refusal [3,7,9]. On the other hand, current studies show that COVID-19-related anxiety and health-related fears are associated with higher vaccine acceptance [16]. Researchers highlight the need to explore different types of fears and anxiety to predict their influence on vaccine acceptance. In this study, we examined how general anxiety, COVID-19-related anxiety, and fear of death are related to the decision to vaccinate against COVID-19.

A systematic review of the literature indicates that most studies reported an association between higher self-esteem and healthier behavior [17]. Preventing disease through vaccination seems like a health-promoting behavior. However, in the case of influenza vaccination, self-esteem is negatively associated with the probability of vaccination [18]. Such results are explained by the tendency of individuals with high self-esteem to ignore disagreeable information and assume that calamities cannot happen, which may lead to risky behaviors such as drinking alcohol or taking drugs [17,19]. Ambiguous results suggest testing whether self-esteem is associated with the decision to vaccinate during the COVID-19 pandemic.

Previous research suggests that understanding the psychological factors responsible for the decision to vaccinate against COVID-19 is crucial to prevent the negative consequences of a pandemic [5,11]. The presented study aimed to examine whether psychological resilience, various types of anxieties, and self-esteem differentiated deciding on SARS-CoV-2 vaccination. We assume that there are differences between vaccinated and unvaccinated individuals. Additionally, we tested the correlation between age and the considered variables.

## 2. Materials and Methods

### 2.1. Participant

The research was conducted on a group of 378 adults (71,3% women), who completed an online survey. The mean age of the respondents was M = 28.37 with SD = 12.73 (range = 18–76 years). Participants were from rural areas (40%), small cities (15%), and urban areas (45%).

### 2.2. Data Collection

The data was collected via online social networking among college students, their family members and friends. A convenience sampling approach was used among the Pedagogical University of Krakow students who were encouraged to tell their family members and friends to sign up to participate in the research. Participants completed the measures anonymously, providing background information about age and gender. Participation in the study was anonymous and confidential.

### 2.3. Measurement

#### 2.3.1. Background Information

Questions related to the background information asked about participants’ age, gender, place of living, vaccination status.

#### 2.3.2. The Trait Anxiety Scale

The Trait Anxiety Scale is a Polish tool (Skala Lęk–Cecha, SL-C) designed by Piksa et al. [20] to measure the intensity of anxiety understood as a personality trait, which is defined as a tendency to perceive situations as dangerous or to expect future events to be threatening, which manifests by characteristic cognitive, affective, behavioral and somatic symptoms. The scale assesses the tendency to perceive a situation as threatening or to predict future events in terms of danger. Such anxiety manifests itself by characteristic symptoms on the cognitive, behavioral, emotional, and somatic levels. The SL-C is a valid one-factor tool and consists of 15 items. Answers are given on a 4-point scale, where the answer “often” = three points, and the answer “never” = zero points. The SL-C score is the sum of all points. The possible scores range from 0 (maximum trait-anxiety intensity) to 45 (minimum trait-anxiety intensity).

#### 2.3.3. Death Anxiety and Fascination

DAFS comprises two scales: death-anxiety and death-fascination [21]. Death anxiety refers to a general fear of death, especially related to oneself. Death fascination contains not only purely cognitive interest in death and dying, but also acceptance of committing suicide and declared death desire. The scale consists of 23 items scored on a scale ranging from 1 (strongly disagree) to 4 (strongly agree).

#### 2.3.4. COVID-19 Fear Scale

FCV-19S consists of 7 items that attempt to measure the fear of COVID-19 [22]. Responding to items on a five-point Likert scale (1 = strongly disagree; 5 = strongly agree), the FCV-19S has been found to be psychometrically sound in assessing fear of COVID-19 in different populations. The higher the score, the greater the fear of COVID-19 among the participants.

#### 2.3.5. The Rosenberg Self-Esteem Scale

SES [23], measures global feelings of self-worth. It is a standardized tool widely known and applied in clinical and research practice. The scale consists of 10 items, 5 expressed in positive statements and 5 in negative statements.

#### 2.3.6. Resilience Evaluation Questionnaire

KOP-26 questionnaire to measure resilience (P) in adults, overall score (P) [24]. The questionnaire defines resilience through three factors: personal (KO), family (RR) and social competence (KS). Personal factors can define a person who can set clear goals and has a purpose in his/her life. She/he believes in her/his abilities and skills. Family factor (RR) describes a person who has close relations with family members. She/he can resolve conflicts and can count on the support of her/his family. She/he also often takes responsibility for completing a task and considers her/his life valuable. The social competence factor defines a person, who easily makes new acquaintances and friends, can win over people, has a large group of friends and quickly adapts to new places. It is easy for her/his to ask for help from other people.

The assessment of the extent to which the respondent agrees with a given statement is made on a 5-point Likert scale (from 1—I completely disagree to 5—I completely agree).

### 2.4. Statistical Analysis

In order to verify the underlying research hypotheses, statistical analysis with R [25] programming language, and JAMOVI [26] statistical software were utilized. Additional packages such as readr [27], psych [28], GPArotation [29], rstatix [30] and corrplot [31] were used to extend the functionality of base R and perform more complex operations.

The reliability of subsequent scales and subscales was checked using Cronbach’s α and McDonald’s ω coefficients. Mean Pearson’s r correlation coefficient between scale items was calculated as well. Then, the normality of distributions of all indicators was assessed with histograms, density plots and qq-plots, whereas indicators were obtained by adding up appropriate items. Subsequently, descriptive statistics were calculated, and Shapiro–Wilk’s normality tests was conducted for continuous variables.

The significance of rank differences between the type of place of residence was evaluated using the H-Kruskal–Wallis test. The significance of mean differences between vaccinated and nonvaccinated people was evaluated using the t-Student test for independent samples. Prior to the proper analysis, the Levene test was performed to check for the homogeneity of variances across the groups. Where necessary, Welch’s correction to Student’s t-test was used. Correlation between all continuous variables is displayed on a corrplot (Figure 1). The global level of significance was assumed at α = 0.050.

## 3. Results

Both Cronbach’s α and McDonald’s ω confirmed very high reliability of obtained results for all scales and subscales. The data from the analysis are presented in Table 1.

According to the histograms, qq-plots, skewness, and kurtosis values, all data were approximately normally distributed. However, probably due to the high sample size, Shapiro–Wilk’s normality tests were in all cases statistically significant. Nevertheless, as the sample sizes in subgroups created by vaccination status were still above 30 and met the central limit theorem criterion, we proceeded with parametric statistical tests. The results of the descriptive statistics analysis are presented in Table 2.

Analysis of rank differences between the type of place of residence for all variables was insignificant.

Next, we calculated the correlation coefficients between the subscales and age and presented these as a corrplot in Figure 1. Under the main diagonal of the matrix, correlation coefficient values were displayed and above the main diagonal, graphical representation of their corresponding magnitude and direction. 

Analysis of Pearson’s r correlation coefficients revealed all statistically significant relationships. A positive correlation was observed between the self-esteem and all resilience subscales ranging from small magnitude with family relation (RR) r = 0.38**, to moderate with personal competence (KO) r = 0.62**. Scales of anxiety were positively correlated with each other; the exception is general anxiety because a higher score means lower anxiety. It is worth noticing, that all variables significantly correlate with age, and only death fascination and personal death anxiety were correlated negatively. The magnitude of this relationship is rather small.

All subscales were examined in respect of differences between subgroups created by the vaccination status. The subjects were divided into two mutually exclusive sets: those who have received at least one COVID-19 vaccination dose, and those who did not receive any vaccination. The results are presented in Table 3.

Significant differences were obtained in terms of three aspects of anxiety: general, against COVID-19, and against death. Effect size estimates for all anxiety variables suggested small magnitudes of differences. It is important to notice that in all aspects of anxiety, higher levels were observed in the vaccinated group. All aspects of resilience and self-esteem were not significantly different.

## 4. Discussion

The current study tests whether general psychological variables are associated with COVID-19 vaccination status. We tested whether vaccinated individuals differ in resilience, self-esteem, and various types of anxiety, from unvaccinated individuals.

We found that those who were vaccinated had significantly higher levels of COVID-19-related anxiety and fear of death, but the severity of general anxiety was lower. Our results seem consistent with previous assumptions that COVID-19-related anxiety and fear of infection and health consequences (perceived severity of illness) are associated with vaccine acceptance [16,32,33]. Vaccination provides a reduction in negative disease outcomes, so people who feel a higher fear of becoming diseased may be more amenable to it. An additional explanation may be that functional fear (of COVID-19) predicts public health compliance during pandemics [34]. Vaccination may be an important aspect of this context because it prevents the spread of the virus and reduces severe disease [35]. Results showing higher levels of general anxiety among unvaccinated individuals may be explained by the relationship between COVID-19 vaccination and psychological distress. Recent analyses reported that the level of mental distress is lower among vaccinated individuals [36]. Additionally, they are also less likely to experience anxiety and worry [37].

In our study, there were no significant differences between the vaccinated and non-vaccinated groups in self-esteem and resilience, although previous studies have suggested that such relations may exist [13,14,17]. To explain these results, it is necessary to distinguish between usual prevention and health decisions and those made during a pandemic. The pandemic situation and the decision to vaccinate are new challenges that trigger new fears among people that cause anxiety [38,39]. New challenges do not necessarily affect relatively stable traits such as self-esteem or resilience. On the other hand, psychological resilience and self-esteem can be certain resources that help to reduce distress and anxiety [40,41]. Our results confirm the direction of these relationships.

Previous research shows a positive relationship between age and fear of COVID-19 [42]. In our study, we observed similar correlations, but it was negligible. Such a result may be caused by the small variation in age among study participants (M = 27.15, SD = 12.69). This result could have been more significant if a larger number of older people had been included in the survey because the elderly might be at higher risk of death and severe complication [43]. We also found a positive correlation between self-esteem and age. This result is consistent with previous results saying that the level of self-esteem increases during adulthood [44].

### Limitations and Further Research Directions

Primarily, our study was exploratory, and it was conducted on a relatively small sample. Only a limited range of sociodemographic variables was included in the analysis, therefore, variables such as education, household situation, and material and professional status should be included in future research. It is also worth noting that people of all age groups should be surveyed, as age can be an important moderator of vaccination. Presented associations and differences could not be interpreted as causal. Regarding different types of anxiety, future studies comparing vaccinated and non-vaccinated should pay attention to the number of doses received and the type of vaccine. Moreover, using other measures of resilience should also be considered.

It is reasonable to expand that quantitative research in the future. We suggest deeper findings by using interviews with respondents or a follow-up study. Such research will allow us to explore the psychological causes of human behaviors during a pandemic.

## 5. Conclusions

There are statistically significant differences between the non-vaccinated and vaccinated group in general anxiety (SLC), COVID-19 anxiety (FCV) and death anxiety (SL). General anxiety was higher in the non-vaccinated group than in the vaccinated group. However, their magnitude is small. There is a medium-positive correlation between age and self-esteem, personal competence, family competence and resilience, respectively.

## Figures and Tables

**Figure 1 ijerph-20-00999-f001:**
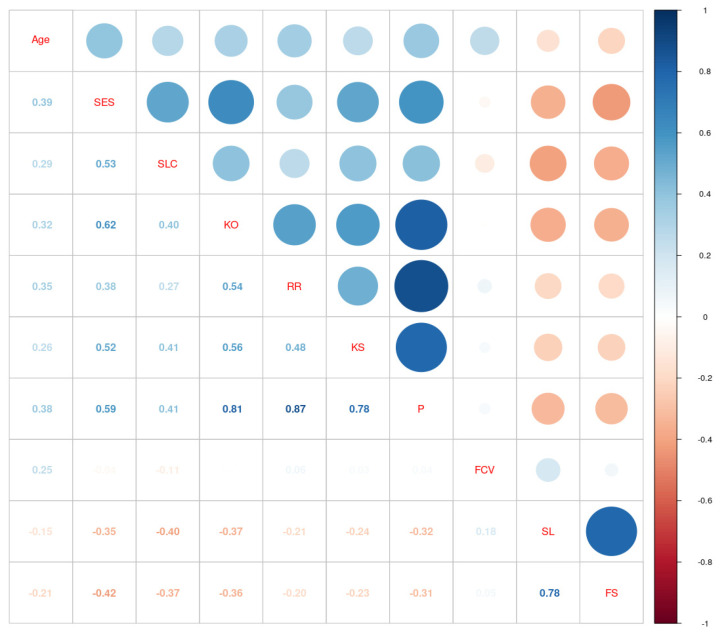
Corrplot illustrating correlations between continuous variables.

**Table 1 ijerph-20-00999-t001:** Reliability analysis.

Scale	Average Pearson’s Correlation	Alpha	Omega
SES	0.44	0.88 (0.87–0.90)	0.89 (0.81–0.92)
SL-C	0.33	0.88 (0.86–0.90)	0.88 (0.84–0.96)
P	0.37	0.94 (0.93–0.95)	0.95 (0.95–0.99)
KO	0.44	0.88 (0.86–0.90)	0.88 (0.86–0.95)
RR	0.55	0.93 (0.92–0.94)	0.93 (0.87–0.95)
KS	0.53	0.87 (0.85–0.89)	0.88 (0.79–0.93)
FCV	0.47	0.85 (0.83–0.87)	0.86 (0.84–0.97)
SLFŚ	0.19	0.85 (0.83–0.87)	0.85 (0.84–0.91)
SL	0.29	0.79 (0.76–0.82)	0.80 (0.74–0.92)
FŚ	0.44	0.92 (0.91–0.93)	0.92 (0.91–0.97)

Note: Data for Alpha and Omega coefficients are presented as coefficient value (95% CI); 95% CI for alpha was calculated according to the Feldt’s method and for omega using bootstrap sampling (100 iterations). SES—self-esteem scale, SL-C—anxiety scale, P—resilience KO—personal competence, RR—family competence, KS—social competence, FCV—COVID-19 anxiety, SLFŚ—death anxiety, SL—personal death anxiety FŚ—death fascination.

**Table 2 ijerph-20-00999-t002:** Descriptive statistics and normality tests for all scales and subscales.

Variable	*M*	*SD*	*Me*	*MAD*	*Min*	*Max*	*Skew.*	*Kurt.*	*W*	*p*
Age	27.15	12.69	20.00	2.97	2.00	76.00	1.34	0.98	0.80	<0.001
SES	27.66	4.88	28.00	4.45	14.00	40.00	−0.01	−0.42	0.99	0.012
SL-C	18.53	7.83	18.00	7.41	2.00	41.00	0.33	−0.38	0.99	<0.001
KO	33.94	7.30	35.00	7.41	13.00	45.00	−0.63	−0.35	0.95	<0.001
RR	42.97	9.49	45.00	8.90	11.00	55.00	−0.90	0.32	0.93	<0.001
KS	18.23	5.96	19.00	7.41	6.00	30.00	−0.19	−0.78	0.98	<0.001
P	91.47	18.23	94.00	19.27	42.00	125.00	−0.52	−0.39	0.97	<0.001
FCV	13.44	5.14	13.00	5.93	7.00	35.00	0.83	0.87	0.98	<0.001
SLC	18.84	4.33	18.00	4.45	9.00	34.00	0.41	0.31	0.96	<0.001
FS	21.17	5.74	20.00	5.93	11.00	41.00	0.72	0.33	0.93	<0.001

Note: M—mean, SD—standard deviation, Me—median, MAD—median average deviation, Min—minimum, Max—maximum, Skew.—skewness, Kurt.—kurtosis, W—Shapiro-Wilk’s test statistic. SES—self-esteem scale, SL-C—anxiety scale, P—resilience KO—personal competence, RR—family competence, KS—social competence, FCV—COVID-19 anxiety, SLFŚ—death anxiety, SL—personal death anxiety FŚ—death fascination.

**Table 3 ijerph-20-00999-t003:** Differences in all subscales in regard to vaccination status.

Dependent Variabls	Not Vaccinated (*n* = 103)	Vaccinated (*n* = 275)	*t*	*p*	Cohen’s *d*	Magnitude
	*M* ± *SD*	*M* ± *SD*				
SESsum	28.34 ± 4.65	27.4 ± 4.95	1.66	0.097	0.19	negligible
SLCsum	20.71 ± 7.70	17.72 ± 7.74	3.35	0.001	0.39	small
KO	34.22 ± 8.05	33.84 ± 7	0.45	0.650	0.05	negligible
RR	43.22 ± 10.21	42.88 ± 9.22	0.32	0.752	0.04	negligible
KS	18.73 ± 5.94	18.04 ± 5.97	1.00	0.319	0.12	negligible
P	92.47 ± 19.99	91.09 ± 17.55	0.65	0.515	0.07	negligible
FCVsum	12.10 ± 4.64	13.94 ± 5.24	−3.14	0.002	−0.37	small
SL	18.09 ± 4.28	19.12 ± 4.32	−2.08	0.039	−0.24	small
FS	20.48 ± 5.85	21.43 ± 5.69	−1.44	0.151	−0.16	negligible

Note: for unequal variances between groups, Welch’s correction was used. SES—self-esteem scale, SL-C—anxiety scale—highest score means lower anxiety, P—resilience KO—personal competence, RR—family competence, KS—social competence, FCV—COVID-19 anxiety, SLFŚ—death anxiety, SL—personal death anxiety FŚ—death fascination.

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
