# Peer review of "The Importance of Resilience and Level of Anxiety in the Process of Making a Decision about SARS-CoV-2 Vaccination"

_ijerph, 2023, doi:10.3390/ijerph20020999_

Round 1

Reviewer 1 Report

·         It would be useful to add what attitudes were in other countries towards vaccination, do they have anything to to with general trust in the system (decision makers).

·         In the Introduction section, it is stated that education and SES were found to be significant, but in description of the sample, these variables were not mentioned.

·         The description of the sample includes the type of place of residence (rural and urban area, small cities), although we do not know beforehand whether it is a significant variable in any way. Also for this current research we do not see a connection.

·         The Discussion section lacks a little deeper explanation of the obtained results, e.g. how they would explain the higher levels of COVID related anxiety and fear of death in vaccinated people.

·         They also state in the Discussion section that results on the positive relationship between age and fear may be caused by the small variation in age among study participants, but that variability was not presented.

Author Response

Dear Reviewer,

Attached you will find our response to review.

Best regards,

Robert Podstawski

Reviewer 2 Report

The research topic of this presentation is timely and appropriate. The method is simple and direct with data collected. There didn't seem to be a follow-up with respondent or an analysis of the data collected. There were no in-depth interviews or follow-up to the findings. This leaves the reader to still ask questions about why or reasons to the psyche of people during the pandemic. What is provided does not answer the question of the psychological make-up of people during the pandemic. The current findings affirms previous studies and really does not add more to those findings. There needs to be further follow-up with larger pool of respondents. The small scope of the research does not move the discussion of the attitudes of people on vaccination further along.  

Author Response

(The authors gave the same response as above.)

Reviewer 3 Report

The authors assessed the psychological determinants of immunization reluctance in depth by measuring the levels of anxiety (death-related and general), fear of COVID-19, self-esteem and resilience among 342 adults. I have the following observations:

1.     The respondents were not selected using any type of accepted sampling – the authors used an online survey that was posted on social media. The sample consists of 71,3% women and is not representative for any socio- demographical category.

2.     The questionnaire uses five scales that the authors do not describe, so there is a lack of continuity between the Measurement part of the article and the Results.

3.     The literature review is superficial and does not quote other relevant studies regarding vaccination.

4.     The authors mention underlying research hypotheses, but do not say what these are (see line 144)

5.     The validity and reliability of the data is questionable at best

6.     The tables are presented as results, but the results are not explained

Author Response

(The authors gave the same response as above.)

Round 2

Reviewer 2 Report

The changes to the text is adequate and improves it from the previous version. The limitations of the research provide the reader a better context to measure the results of the findings. This also open the possibility for future research and an expansion of the findings. It sounds like it is the desire of the researcher to do further study of this and this reviewer encourage such efforts. 

Author Response

Dear Reviewer,

I kindly inform you that the English correction has been done.

Best regards,

Author